# Genes associated with body weight gain and feed intake identified by meta-analysis of the mesenteric fat from crossbred beef steers

**Amanda K. Lindholm-Perry**⬤*, **Harvey C. Freetly, William T. Oliver, Lea A. Rempel, Brittney N. Keel**

Agricultural Research Service, United States Department of Agriculture, United States Meat Animal Research Center, Clay Center, Nebraska, United States of America

* amanda.lindholm@usda.gov

**Data Availability Statement:** Sequence data is available in NCBI Sequence Read Archive under accession number PRJNA528040.

## Abstract

Mesenteric fat is a visceral fat depot that increases with cattle maturity and can be influenced by diet. There may be a relationship between the accumulation of mesenteric fat and feed efficiency in beef cattle. The purpose of this study was to identify genes that may be differentially expressed in steers with high and low BW gain and feed intake. RNA-Seq was used to evaluate the transcript abundance of genes in the mesenteric fat from a total of 78 steers collected over 5 different cohorts. A meta-analysis was used to identify genes involved with gain, feed intake or the interaction of both phenotypes. The interaction analysis identified 11 genes as differentially expressed. For the main effect of gain, a total of 87 differentially expressed genes (DEG) were identified ($P_{ADJ}$<0.05), and 24 were identified in the analysis for feed intake. Genes identified for gain were involved in functions and pathways including lipid metabolism, stress response/protein folding, cell proliferation/growth, axon guidance and inflammation. The genes for feed intake did not cluster into pathways, but some of the DEG for intake had functions related to inflammation, immunity, and/or signal transduction (*JCHAIN*, *RIPK1*, *LY86*, *SPP1*, *LYZ*, *CD5*, *CD53*, *SRPX*, and *NF2*). At $P_{ADJ}$<0.1, only 4 genes (*OLFML3*, *LOC100300716*, *MRPL15*, and *PUS10*) were identified as differentially expressed in two or more cohorts, highlighting the importance of evaluating the transcriptome of more than one group of animals and incorporating a meta-analysis. This meta-analysis has produced many mesenteric fat DEG that may be contributing to gain and feed intake in cattle.

## Introduction

The most significant cost in beef cattle production is feed, which accounts for 60–80% of the total cost of production [1]. Improvements to the feed efficiency of beef cattle is a means by which costs can be reduced. In addition, it is also important to ensure that the feed is efficiently converted into meat products for consumption. Mesenteric fat is a visceral adipose depot located around the intestines that can be affected by diet. Cattle diets with high fat and starch

**Funding:** The authors received no specific funding for this work.

**Competing interests:** The authors have declared that no competing interests exist.

leads to an accumulation of mesenteric fat [2]. In chickens, visceral fat has no economic value and is considered a source of feed inefficiency with more feed efficient chickens having lower abdominal fat [3]. This is also presumably the case for other livestock species, like cattle and swine. Because there may be a relationship between abdominal and visceral fat depots and feed efficiency in cattle; thus, we hypothesized that there may be differences in the mesenteric fat gene expression of feedlot steers with high and low gain and high and low feed intake.

A previous study using microarray analysis on the mesenteric fat of 16 of the steers in this study identified genes involved in proteolysis, transcription, transport, immune function, glycerol degradation and oxidative stress [4]. The current study was performed on five cohorts of steers, including the 16 from Lindholm-Perry et al. [4], and a meta-analysis was used to identify genes involved in gain and feed intake across cattle of various breeds and seasonal conditions. The genes presented here may be more robust across populations of beef cattle than those identified in only one cohort of animals.

A previous study performed on the muscle tissue of these same animals illustrated that there is significant variation in differentially expressed genes identified by cohort [5]. This may be due to various environmental and genetic differences represented by each cohort. However, the overarching goal of these types of transcriptome experiments is to identify genes that are contributing to the phenotype of the species being studied. One way to isolate the genes that contribute to phenotype, regardless of environment, is to evaluate several cohorts of animals and analyze the data using a meta-analysis [5]. These types of analyses are more likely to produce valid results in independent datasets [6–8].

The objective of this study was to determine whether differentially expressed genes in the mesenteric fat were associated with variation in gain and feed intake in five cohorts of beef steers. We hypothesized that the expression of genes in pathways such as lipid metabolism, protein turnover and oxidative stress would vary by phenotype. However, we designed this as a discovery study using RNA sequencing to allow us to identify all of the genes that were differentially expressed, rather than targeting specific candidate genes.

## Materials and methods

### Animal care and use

The U.S. Meat Animal Research Center (USMARC) Animal Care and Use Committee reviewed and approved all animal procedures. The procedures for handling cattle complied with the Guide for the Care and Use of Agricultural Animals in Agricultural Research and Teaching [9].

### Cattle population

Steers (n = 80) from the USMARC Germplasm Evaluation project [10], were selected for this study over 5 cohorts. This population of cattle includes the common U.S. beef breeds: Angus, Beefmaster, Brahman, Brangus, Brown Swiss, Charolais, Chiangus, Gelbvieh, Hereford, Limousin, Maine Anjou, Red Angus, Romosinuano, Bonsmara, South Devon, Salers, Santa Gertrudis, Shorthorn, and Simmental. At age 350 ± 54 days, steers were placed on a corn-based finishing diet *ad libitum* feeding trial for 64–92 days. Steers from the Spring 2012 study were fed for 64 days: 57.57% high moisture corn, 8% ground alfalfa hay, 30% wet distiller's grains with solubles (WDGS), and 4.25% Steakmaker (protein supplement, Land O'Lakes, Arden Hills, MN, USA) as dry matter. Steers from the Fall 2012 and Spring 2013 studies were fed for 92 and 64 days, respectively: 57.35% dry rolled corn, 8% ground alfalfa hay, 30% WDGS, 4.25% Steakmaker and 0.4% urea as dry matter. Steers from the Fall 2013 and Fall 2014 studies were fed for 84 and 78 days, respectively: 57.35% dry rolled corn, 8% ground alfalfa hay, 30%

WDGS, 4.25% Steakmaker with Tylan (Tylan manufactured by Elanco Animal Health, Greenfield, IN, USA) and 0.4% urea as dry matter. All diets were formulated to meet or exceed nutrient requirements to express maximum growth potential [11]. Steers were housed in pens with 8 feed bunks with Insentec Roughage Intake Control Feeding Systems (Insentec B.V., Marknesse, The Netherlands) to measure feed intake. The selection of steers, 16 animals from each of 5 cohorts, was previously described [5]. Animals selected for this study were given *ad libitum* access to the same diet and water until slaughter. On each day of slaughter, one steer from each of the 4 phenotypic groups (n = 4) was stunned with captive bolt and all animals were processed within 3 hours. Length of time between the feed trial and slaughter varied slightly by cohort: cohort 1 was 12–18 days; cohort 2 was 19–22 days; cohort 3 was 5–8 days; cohort 4 was 20–28 days; and cohort 5 was 11–14 days. Steers were selected for the study as follows: body weight gain and feed intake were measured on 5 cohorts of steers from 2012 to 2014. Steer weights were taken on the first and last two days of the study and once every three weeks throughout the study. Total body weight gain was calculated by regressing body weight on days on study using a quadratic polynomial and average daily gain (ADG) was calculated by dividing total body weight gained by days on study. Average daily dry matter intake (ADDMI) was calculated by dividing the total feed intake by days on study. Sixteen steers were selected from each cohort by ranking them based on their bivariate mean (of ADG and ADDMI) and 4 animals with the greatest deviation from the mean in each Cartesian quadrant were sampled. Summary statistics for gain and feed intake by cohort is presented in Keel et al. [5]. Breed composition by phenotypic group and medical and health records were also evaluated. Phenotypes were not confounded [5].

## Tissue collection and RNA isolation

On each day of slaughter, the four animals selected were processed serially within a three-hour time frame. The tissue sample was obtained from each steer within approximately 30–40 minutes after exsanguination. Mesenteric fat was removed from approximately 1-m distal to the duodenum. Mesenteric fat samples were diced into 50–100 mg pieces and flash frozen in liquid nitrogen then stored at -80˚C until they could be processed further. Tissue was homogenized in TriPure reagent (Cat # 11667165001; Roche, Indianapolis, IN, USA). RNA was isolated according to the manufacturer's protocol with an extra 20-minute centrifugation following the addition of chloroform. The resulting RNA pellets were further processed using RNeasy Plus Mini Kit columns (Cat # 74136; Qiagen, Valencia, CA, USA) according to the manufacturer's instructions to remove genomic DNA.

The concentration of the RNA was determined with a Nanodrop 8000 spectrophotometer (Thermo Scientific, Wilmington, DE, USA). The 260/280 ratios were ≥ 1.8 and a subset of 20 samples (at least 3 samples from each cohort) were analyzed for quality on either the Agilent Bioanalyzer 2100 or the Agilent Tapestation 2200 (Cat # 5067-1511 and 5067–5576; Agilent Technologies, Santa Clara, CA, USA) and produced an average RIN of 8.2 with a range of 7.1 to 9.2.

## RNA sequencing

Samples were prepared for RNA sequencing with the Illumina TruSeq Stranded mRNA High Throughput Sample kit and protocol (Cat # RS-122-2103; Illumina Inc., San Diego, CA, USA). The libraries were diluted to 4nM (Teknova, Hollister, CA. USA). One sample failed to produce a library. The remaining 79 samples were paired-end sequenced with 150 cycle high output sequencing kits for the Illumina NextSeq. Sequence data is available in NCBI Sequence Read Archive under accession number PRJNA528040.

## Processing RNA-seq data

The quality of the raw paired-end sequence reads in individual fastq files was assessed using FastQC (Version 0.11.5; www.bioinformatics.babraham.ac.uk/projects/fastqc), then reads were trimmed to remove adapter sequences and low-quality bases using the Trimmomatic software (Version 0.35) [12]. The remaining reads were mapped to the newly released ARS-UCD1.2 genome assembly (Genbank Accession GCA_002263795.2) using Hisat2 (Version 2.1.0) [13], and the NCBI annotation for ARS-UCD1.2 (Release 106) was used to guide the alignment. Stringtie [14] was used to determine read counts for each of the 34,624 annotated genes in the ARS-UCD1.2 genome assembly. Genes with low read counts were filtered out of the dataset when there were < 15 reads in at least 16 samples. This produced a set of 16,701 genes for downstream analyses. One library was removed from the analysis due to low read counts and lower mapping percentage (n = 78).

## Meta-analysis of differential gene expression

The meta-analysis procedure developed in Keel et al. [5] was used to identify differentially expressed genes associated with gain and feed intake. Briefly, cohorts were analyzed separately using the DESeq2 package [15] with the following generalized linear model (GLM):

$$Y = Gain + Intake + Gain \; x \; Intake \qquad (1)$$

Raw P-values from each experiment were combined into one test statistic via Fisher's method [16], and the Benjamini-Hochberg method [17] was applied to the raw meta P-values to correct for multiple testing. Genes with adjusted meta $P$-value $\leq 0.05$ were considered statistically significant. DEGs associated with a main effect and also the interaction term were excluded from the main effect list of DEGs since this indicates that the main effect is dependent on the interaction term.

## Jackknife reproducibility analysis

Jackknife sensitivity analysis was employed to evaluate the robustness of the results. The meta-analysis procedure was repeated multiple times, each time with removal of a single cohort from the baseline group of cohorts [18].

## Functional and pathway analysis of DEGs

Over-represented gene functions and pathways of over-represented DEG ($P_{ADJ}<0.1$) were determined using the PANTHER classification system (Version 13.1) [19] and the Database for Annotation, Visualization and Integrated Discovery (DAVID; Version 6.8) [20,21]. Enrichment analysis of gene function was performed using PANTHER's implementation of the binomial test of overrepresentation. Significance of gene ontology (GO) terms was assessed using the binomial test of overrepresentation and the default Ensembl *Bos taurus* GO annotation as background for the enrichment analysis. The default parameters for the KEGG pathway and gene function analyses in DAVID were used with the official gene symbols and *Bos taurus* annotation and to evaluate genes that were over-represented in the list of DEG ($P_{ADJ}<0.1$).

## QIAGEN Ingenuity® Pathway Analysis

Ingenuity® Pathway Analysis (IPA®, QIAGEN Redwood City, CA; www.qiagen.com/ingenuity) was used to deduce direct and indirect molecular relationships among differentially expressed genes for main effects of gain and intake, and gain x intake interaction. Each of the data sets was imported with a Flexible Format using Gene symbol as the identifier. A core

analysis was performed on genes in each set, where a P-value for each network is calculated according to the fit of the user's set of significant genes and the size of the network.

## Results

### Sequencing statistics

RNA-Seq libraries from the mesenteric fat tissue of 78 steers with high and low ADG and ADDMI were sequenced. Over 4.5 billion 75-bp paired-end reads were generated, with an average of 58.3 million reads per sample. After trimming adapter sequences and low quality bases, the resulting high quality reads were mapped to the *Bos taurus* ARS-UCD1.2 genome assembly with an average 97.1% read mapping rate.

### Meta-analysis of DEGs associated with gain and feed intake across cohorts

Average daily gain and ADDMI for each cohort is presented in Keel et al. [5]. After multiple testing correction, we identified 87 DEG for the main effect of gain, 24 genes for the main effect of feed intake, and 11 DEG for the gain x intake interaction (S1–S3 Tables). Significant genes were inspected for consistency, defined as having the same log-fold change direction across all 5 cohorts. We found that two significant genes (*RAB7B* and *CHAC1*) were consistent for direction of expression in all cohorts for the gain main effect, while no DEGs were consistent for the intake main effect or the gain x intake interaction.

### Jackknife analysis

Robustness of the results were assessed using a jackknife sensitivity analysis, where for each term in the model, five separate meta-analyses were performed, each omitting a single cohort. The results are shown in S4–S6 Tables. For the gain main effect, the jackknife analyses produced similar numbers of DEGs to that of the original meta-analysis (87 DEGs): 109, 89, 52, 49, and 67 DEGs for the jackknife analysis that removed Cohort 1, 2, 3, 4, or 5; respectively (Jacknife $P < 0.05$ in S4 Table). The number of DEGs identified in the jackknife analyses for the intake main effect varied more than for gain, with 46, 42, 17, 5, and 44 DEGs for the jackknife analysis that removed Cohort 1, 2, 3, 4, or 5; respectively (Jacknife $P < 0.05$ in S5 Table). The jackknife analyses for the interaction effect identified 20, 15, 8, 4, and 7 DEGs for the jackknife analysis that removed Cohort 1, 2, 3, 4, or 5; respectively (Jacknife $P < 0.05$ in S6 Table), which was similar to the number of DEGs identified in the full meta-analysis (11 DEGs).

For the gain main effect, there was one DEG, *OLFML3*, that was robust enough to pass all five jackknife analyses. Thirty-one DEGs failed only 1 jackknife analysis, while 32, 14, 9, and 0 DEGs failed to pass 2, 3, 4, and 5 jackknife analyses, respectively. For the intake main effect, there were no DEGs that passed or failed all five jackknife analyses. Eleven DEGs failed only 1 jackknife analysis, while 9, 2, and 1 DEGs failed to pass 2, 3, and 4 jackknife analyses, respectively. For the interaction effect, there was one DEG, *LOC100300716*, that passed all five tests and none failed all five. A total of 1, 8, 0, and 1 DEGs failed to pass 1, 2, 3, and 4 tests, respectively.

### Functions of DEG

PANTHER gene ontology analysis of the DEGs indicated that genes significant for the gain main effect were involved in catalytic activity (38.9%), binding (29.2%), metabolic process (18.1%), localization (15.3%), and response to stimulus (12.5%). No GO terms were significantly over- or under-represented in this gene set.

**Table 1. Panther pathways for Gain DEG.**

| Pathway | # | expected | Fold Enrichment | +/- | P value | FDR |
|---|---|---|---|---|---|---|
| Axon guidance mediated by Slit/Robo | 4 | 0.12 | 32.60 | + | 1.13E-05 | 9.26E-04 |
| Axon guidance mediated by netrin | 4 | 0.17 | 23.17 | + | 3.87E-05 | 2.11E-03 |
| Cytoskeletal regulation by Rho GTPase | 5 | 0.42 | 11.83 | + | 8.33E-05 | 3.42E-03 |
| Inflammation mediated by chemokine and cytokine signaling pathway | 9 | 1.26 | 7.15 | + | 6.09E-06 | 9.98E-04 |

Pathway analysis in PANTHER produced four pathways over-represented in the list of DEG ($P < 0.05$; Table 1). These included: axon guidance mediated by Slit/Robo, axon guidance by netrin, cytoskeletal regulation by Rho GTPase, and inflammation mediated by chemokine and cytokine signaling pathway. The DAVID pathway analysis also identified axon guidance as over-represented in the list of DEG ($P < 0.05$; Table 2). In addition, two other pathways (MAPK signaling and protein processing in the endoplasmic reticulum) approached significance ($P < 0.1$).

Similar to the gain main effect genes, genes significant for the intake main effect were involved in catalytic activity (50%), binding (30%), metabolic process (33.3%), and response to stimulus (6.7%). Again, no GO terms were identified by PANTHER to be significantly over- or under-represented in this set. Pathways identified by DAVID as over-represented from the list of DEG genes for intake were; hematopoietic cell lineage and Toll-like receptor signaling pathway ($P < 0.1$; Table 3).

### Ingenuity Pathway Analysis (IPA)

The top molecular and cellular functions represented in the list of DEG for gain included cellular development, cellular growth and proliferation, cellular assembly and organization, cellular function and maintenance, and lipid metabolism (Table 4). The genes involved in lipid metabolism, molecular transport and small molecule biochemistry included *ACLY, AKR1B1, BAG3, GALR3, KCNMA3, LIPG, LPIN1, MAPT, NEIL1, PTGS2, TARDBP*. Genes involved in cellular development, growth/proliferation, and function/maintenance included *ACLY, AKR1B1, ARHGEF26, CDKN2B, DLX3, HAS3, HSP1A1, HSPA1B, LDB3, MAP2K6, MAPT, MYLK, NINL, PTGS2, SFRP1, STTG3B, TARDBP, UNC5A, VEGFC*. Canonical pathways identified in the list of DEG for gain included ErbB signaling, cholecystokinin/gastrin-mediated signaling, agrin interactions at neuromuscular junctions, role of MAPK signaling in the pathogenesis of influenza, and IL-17 signaling. The genes that were responsible for these pathways were a combination of *MAP2K6, NRG2, PAK5, PTGS2, RAP1A* and *RHOH*.

The molecular and cellular functions over-represented in the list of DEG for intake included small molecule biochemistry, cellular compromise, cell morphology; cell signaling and post-translational modification. Genes that were included in more than one of these functions were: *NF2, RIPK1, SPP1, CD5, LYZ, CPM, PRMT7*, and *B3GAT1*.

### Discussion

To date, there have been many transcriptome studies to identify differentially expressed genes related to feed efficiency from various tissues in livestock; however, mesenteric fat is largely

**Table 2. DAVID pathways for main effect of gain.**

| Term | Count | P value | Genes |
|---|---|---|---|
| Axon guidance | 4 | 0.040 | UNC5A, PAK5, NTNG1, SLIT1 |
| MAPK signaling | 5 | 0.069 | MAPT, RAP1A, HSPA1A, MAP2K6 |
| Protein processing in endoplasmic reticulum | 4 | 0.083 | STT3B, HSPH1, HSPA1A |

**Table 3. DAVID pathways for main effect of intake.**

| Term | Count | P value | Genes |
|---|---|---|---|
| Hematopoietic cell lineage | 2 | 0.08 | LOC782367, CD5 |
| Toll-like receptor signaling pathway | 2 | 0.09 | RIPK1, SPP1 |

under-represented in these studies. The goal of our study was to identify differentially expressed genes in the mesenteric fat that are associated with beef steer body weight gain and feed intake across several contemporary groups of animals. To do this, we used a meta-analysis method to detect differentially expressed genes from 5 cohorts of animals. This allowed us to identify several genes among the 80 animals tested that may be involved in gain and feed intake. Although we corrected the meta-analysis P-values and only reported the genes as differentially expressed with $P_{ADJ} < 0.05$, this still implies that 5% of the significant tests are false positives.

This study evaluated the mesenteric fat transcriptome of 5 cohorts of steers, which produced differentially expressed genes for gain, feed intake and their interaction; however, the level of transcription is only one of many components that may be affecting phenotype. In addition, the lack of agreement of DEG from one cohort to another illustrates the importance of evaluating the transcriptome in more than one group or population of animals, especially those like cattle that are exposed to environmental variables such as weather, management, and subtle changes in feed ingredients. We hypothesized that we would identify genes involved in protein turnover, lipid metabolism and oxidative stress in the mesenteric fat of these steers. While we did identify some genes functioning within some of these pathways, they were not as numerous as we expected. This may be due, in part, to other factors that affect gene expression, such as epi-genetic modification, chromatin structure, nucleosomes and nucleic acid post transcriptional remodeling, miRNA abundance, long non-coding RNA. In addition, there may be post-transcriptional, translational modifications, and enzyme modifications that may be affecting protein turnover in these steers. Thus, while this study focuses on the

**Table 4. Significant molecular and cellular functions for DEGs associated with the gain main effect identified using IPA.**

| Canonical Pathway | P value | # Genes | Genes |
|---|---|---|---|
| Cellular development | 0.012 | 24 | ACLY, AKR1B1, ARHGEF26, CDKN2B, DLX3, HAS3, HSPA1A, HSPA1B, KCNMA1, LDB3, LPIN1, MAP2K6, MAPT, MYLK, NINL, NRG2, PTGS2, RHOH, SFRP1, SH2B2, STT3B, TARDBP, UNC5A, VEGFC |
| Cellular growth and proliferation | 0.012 | 19 | ACLY, AKR1B1, ARHGEF26, CDKN2B, DLX3, HAS3, HSP1A1, HSPA1B, LDB3, MAP2K6, MAPT, MYLK, NINL, PTGS2, SFRP1, STTG3B, TARDBP, UNC5A, VEGFC |
| Cellular assembly and organization | 0.012 | 24 | BAG3, CHMP1B, CNTN4, HAS3, HSPA1A, HSPA1B, LDB3, LPIN1, MAP2K6, MAPT, MYLK, NINL, NTNG1, PAK5, PTGS2, RAP1A, SFRP1, SGK2, SH2B2, SHROOM3, SLIT1, TARDBP, UNC5A, VEGFC |
| Cellular function and maintenance | 0.012 | 35 | ACLY, AKR1B1, ARHGEF26, BAG3, CCL1, CCL8, CD5, CHMP1B, CNTN4, HAS3, KCNMA1, LDB3, LIPG, LPIN1, MAP2K6, MAPT, MYLK, NINL, NRG2, NTNG1, PAK5, PDE4B, PTGS2, RAP1A, RHOH, SFRP1, SGK2, SH2B2, SHROOM3, SLIT1, SRPX, STT3B, TARDBP, UNC5A, VEGFC |
| Lipid metabolism | 0.013 | 11 | ACLY, AKR1B1, BAG3, GALR3, KCNMA1, LIPG, LPIN1, MAPT, NEIL1, PTGS2, TARDBP |

transcriptome, it is important to consider that there are likely many other factors involved in these phenotypes.

## Genes/pathways identified for gain

Differentially expressed genes with the same direction of expression over all of the cohorts may be the most robust and straightforward to validate among other cattle populations. While 87 genes were identified as differentially expressed for the main effect of gain, only 2 (*CHAC1* and *RAB7B*) displayed the same direction of expression over all 5 cohorts of animals. One of the genes (*CHAC1*) has functions related to oxidative stress. The *CHAC1* gene specifically cleaves glutathione and was down-regulated in animals with lower gain in the current study. The overexpression of *CHAC1* in cell culture was shown to degrade cellular glutathione suggesting a feedback mechanism that sensitizes cells to oxidative stress [22]. The animals with higher gains may be mitigating the effects of oxidative stress by reducing cellular levels of glutathione. Genes involved in oxidative stress have been previously identified in other tissues from these same animals [4, 5, 23, 24]. A specific example of this includes the *UQCRQ* gene, which was identified as differentially expressed in the muscle tissue from these animals and displayed the same direction of expression in all cohorts [5]. Oxidative stress has been proposed in many other studies as a factor that affects feed efficiency in swine, poultry, and beef cattle [25–30].

The current study also produced several DEG for gain that failed 0 or 1 jackknife test. These genes included: *OlFML3*, *GCAT*, *EBPL*, *HSPH1*, *MOSPD1*, *ADGRL4*, *LOC782367*, *SFRP1*, *GALR3*, *RHOH*, *HPS6*, *CDKN2B*, *MGC152281*, *SMUG1*, *SRPX*, *RAB7B*, *GPT*, *CCDC117*, and *MAPT*. A gene that may be of particular interest is *GALR3*. GALR3 is one of 3 receptors that bind the galanin neuropeptide. Galanin has been shown to modulate physiologic processes including hormone secretion and feeding behavior. Mice without the *GALR3* gene had reduced psoriasis symptoms as a result of delayed angiogenesis, reduced neutrophil infiltration, and lower levels of cytokines [31]. In the steers evaluated in the current study, expression of *GALR3* was lower in the animals with higher gain. A reduction in neovascularization and inflammation in the mesenteric fat stores may allow for improvements in weight gain. Inflammation is an energetically expensive process and is negatively associated with growth in humans [32], and pigs [33]; thus, cattle with a reduction in the production of cytokines may be able to devote more energy towards gain.

For the main effect of gain, several genes involved in lipid metabolism (*ACLY*, *AKR1B1*, *BAG3*, *GALR3*, *KCNMA1*, *LIPG*, *LPIN1*, *MAPT*, *NEIL1*, *PTGS2*, *TARDBP*) were identified as differentially expressed, as we would expect for an adipose depot. Another notable cluster of genes differentially expressed included three stress response heat shock protein genes; *HSPB3*, *HSPA1A*, and *HSPH1*; which all mirrored direction of expression by cohort, except in cohort 4 where *HSPA1A* was up-regulated in animals with higher gain and the other two were down-regulated. Heat shock proteins can rescue unfolded proteins, and increased expression of *HSPH1* (Hsp110) and *HSPA1A* (Hsp72) can be a result of physiological or cellular stress [34, 35]. Heat shock proteins have also been previously identified as differentially expressed in the small intestine and spleen of one cohort of the animals from the current study [24,36], and also in the liver and duodenum of pigs divergent for residual feed intake [37]. In the current study, *HSPH1*, was expressed in lower abundance in animals with high weight gain in 4 of the 5 cohorts. Further evidence of potential cellular stress was the differential expression of two genes identified with functions in glutathione metabolism, *CHAC1* and *GSTT2*. *GSTT2*, a gene associated with oxidative stress, has been shown to respond *in vitro* to paraoxon, an agent that increases superoxide accumulation in cells [38]. In dogs, the expression of *GSTT2* increased in

the adipose tissue as feed intake increased [39]. Thus, Grant et al. [39] proposed that its expression may be related to metabolic stress during adipose tissue expansion. *CHAC1* was found in lower transcript abundance in animals with low gain in all cohorts and *GSTT2* displayed lower transcript abundance in 3 of the 5 cohorts. Assuming that cattle with high gains on a feedlot diet, are gaining lean tissue rather than accumulating fat, these data might suggest that less inflammation and a more effective clearance of glutathione and elevated reactive oxygen species reduces the need for heat shock proteins in the mesenteric fat of the higher gaining animals.

Axon guidance pathways were identified by DAVID as those that were enriched by differentially expressed genes for gain (*UNC5A*, *PAK5*, *NTNG1*, *SLIT1*). Three of these genes (UNC5A, NTNG1 and SLIT1) interact with each other for axon or neurite growth. White adipose tissue, of which mesenteric fat is comprised, is innervated by both the sympathetic and parasympathetic nervous system [40], which indicates a balance between the sympathetic "fight, flight or freeze" nervous system and the "feed and breed" and "rest and digest" parasympathetic nervous system [40]. This balance may be affected by the expression of these axon guidance genes. Visceral adipose tissue has been shown to have more efferent sympathetic axons and capillaries than subcutaneous fat [41]. Moreover, neurons from the spinal cord projecting into the visceral fat are different than those projecting into the subcutaneous fat, suggesting that these two fat depots have unique actions on, or in response to, the central nervous system.

## Genes/pathways identified for intake

The genes identified as differentially expressed for the main effect of intake did not cluster into functional gene categories or pathways well. In fact, only trends towards significance for two pathways, hematopoietic cell lineage and Toll-like receptor signaling pathway, were detected. Based on literature searches of the functions of each of the DEG for intake, several appeared to have roles in inflammation or immune function (*JCHAIN*, *RIPK1*, *SPP1*, *CD5*, *CD53*, *LYZ*, *LOC782367*, and *LY86*). Four of these genes (*JCHAIN*, *RIPK1*, *LOC782367*, and *LY86*) were among the most robust genes failing none or only one of the jackknife tests. The number of immune and inflammatory response genes identified for both gain and intake in steers in the mesenteric fat depot may not be surprising considering that this tissue is often referred to as an endocrine organ [42,43]. Visceral fat is a unique type of adipose tissue that sequesters polyunsaturated fatty acids necessary for immune functions, like lymph node activity, and that mesenteric fat responds to immune and inflammatory signals [44, 45].

## Genes identified for the gain x intake interaction

Two genes identified as differentially expressed for the interaction term passed at least 4 jackknife tests. These were *LOC100300716* and *PTAFR*. The *LOC100300716* gene is the immunoglobulin heavy variable 4-38-2 that is involved in antigen recognition and is conserved in human and other primates. The PTAFR receptor binds platelet activating factor (PAF), which has been implicated in a number of diseases and disorders like asthma, allergies and is also a lipid mediator of inflammatory responses [46]. These genes underscore the potential involvement that immune responses and inflammatory pathways seem to have with feed efficiency in the mesenteric adipose tissue. Other genes that were identified in the gain x intake interaction analysis of interest were *AACS*, *H19* and *IL18BP*. The AACS gene was identified as a member of the *INSIG1* hub of genes involved in residual feed intake in cattle [47]. The genes in this network are involved in energy, lipid and steroid metabolism, as all of which have been linked to feed efficiency [48, 49]. Interestingly, *INSIG1* was identified as differentially expressed in the current study, and also in a previous study of the muscle tissue of these same animals [5].

Acetyl-CoA synthetase (AACS) enzyme is required for the conversion of ketone bodies into energy within adipose tissue. *AACS* transcripts are highly expressed in rodent visceral fat [50] and become up-regulated during adipocyte differentiation, utilizing ketone bodies for lipogenesis. *H19* is located in an imprinted chromosomal region and encodes a long non-coding RNA. It is thought to regulate gene expression and may also have a role in negatively regulating body weight and cell proliferation [51]. Lastly, the protein encoded by *IL18BP* functions as an inhibitor of the proinflammatory cytokine, IL18. It binds IL18, prevents the binding of IL18 to its receptor, and thus inhibits IL18-induced IFN-gamma production, resulting in reduced T-helper type 1 immune responses. The *IL18BP* gene was down-regulated in the low gain animals in 2 cohorts with significant p-values (S4 Table).

## Comparison of adipose and muscle DEG

The muscle tissue from these same animals was also analyzed in a meta-analysis [5]. There were four genes that were identified as differentially expressed in the both of these meta-analyses ($P_{corrected} < 0.1$). They were *CIDEA*, *DEDD2*, *H19* and *INSIG1*. Interestingly, CIDEA knockout mice are resistant to diet-induced obesity [52]. Indeed, *CIDEA* expression was lower in the mesenteric fat of the low gain cattle in 4 of the 5 cohorts in the current study. The death effector domain containing 2 (*DEDD2*) gene was one of 12 candidate genes identified in an intra-module in the liver and duodenum of cattle with variation in residual feed intake [37]. In addition, knockout mice for DEDD maintained normal concentrations of blood glucose, hormones, and growth factors in comparison to wild-type mice; however, their body size was reduced indicating a possible alteration to the cellular or physiological mechanisms of metabolism [53]. The *INSIG1* gene has been identified as a candidate gene in many GWAS and transcriptome studies designed to evaluate feed efficiency in cattle [47, 54, 55] and chickens [3, 56]. Expression of INSIG1 increased in fat tissue of normal mice constitutively with diet-induced obesity [57]. Li and others concluded that INSIG1 functions as a feedback mechanism to control lipogenesis and adipogenesis. In contrast, Graugnard and others [55] determined the activity of INSIG1 in cattle muscle was to initiate intramuscular fat deposition. These contradicting results may be species affiliated or rather tissue specific, murine epididymal fat pads, a visceral fat depot, versus longissimus lumborum of cattle.

## Comparison with previous microarray study of one cohort

The transcriptome of one of the 5 cohorts of animals was previously assessed using microarray technology [4]. The DEG identified in that study were primarily associated with animals with the low gain/high intake phenotype. While some of the genes identified in that study were also identified within this cohort in the current study, they did not maintain significance in the meta-analysis. Conversely, a total of eight genes (*CDKN2B*, *CHMP1B*, *CHAC1*, *DLX3*, *SLIT1*, *MOSPD1*, *ANTXRL*, and *B3GAT1*) were identified as differentially expressed for gain or intake in the current meta-analysis and were also nominally significant in the previously performed microarray study [4]. However, it is important to note that none of the genes passed FDR correction in the microarray study, and therefore would not have been selected as critical drivers of gain or intake from the previous study. These results highlight the importance of evaluating more than one group of animals in which thousands of genes are being evaluated for specific phenotypes.

## Conclusions

The current study, using 5 cohorts of steers in conjunction with a meta-analysis, has produced a number of highly specialized and functional candidate genes that may be responsible for

gain and feed intake in cattle. Genes clustering into lipid metabolism, cellular development, growth/proliferation biological functions and inflammatory and protein processing pathways were identified. Many genes were identified that passed 4–5 jackknife analyses, as well as two DEG with the same direction of expression across all cohorts and four that were also identified in the muscle tissue of these sample animals [5]. These may be promising candidate genes to validate for gain and feed intake phenotypes in additional populations of animals.

## Supporting information

**S1 Table. DEGs associated with the gain main effect in the individual cohort and meta-analyses.** Genes are ordered by adjusted meta-P-value. The individual cohort cells for DEGs identified in the meta-analysis are colored according to the sign of their log2 fold change, where green indicates up-regulation and red indicates down-regulation in high gain. Genes with all gray cell indicate those that were excluded because they were also significant for the gain by intake interaction term.
(PDF)

**S2 Table. DEGs associated with the intake main effect in the individual cohort and meta-analyses.** Genes are ordered by adjusted meta-P-value. The individual cohort cells for DEGs identified in the meta-analysis are colored according to the sign of their log2 fold change, where green indicates up-regulation and red indicates down-regulation in high intake. Genes with all gray cell indicate those that were excluded because they were also significant for the gain by intake interaction term.
(PDF)

**S3 Table. DEGs associated with the gain by intake interaction effect in the individual cohort and meta-analyses.** Genes are ordered by adjusted meta-P-value. The individual cohort cells for DEGs identified in the meta-analysis are colored according to the sign of their log2 fold change, where green indicates up-regulation and red indicates down-regulation in low gain, low intake.
(PDF)

**S4 Table. Jackknife sensitivity analysis results for the DEGs associated with the gain main effect.** Genes with all gray cell indicate those that were excluded because they were also significant for the gain by intake interaction term. Jackknife 1 P-value gives the adjusted P-value for the meta-analysis with Cohort 1 removed, Jackknife 2 P-value gives the adjusted P-value for the meta-analysis with Cohort 2 removed, and so on. Yellow cells indicate jackknife analyses where the P-value was insignificant, i.e. the gene failed to pass the jackknife analysis.
(PDF)

**S5 Table. Jackknife sensitivity analysis results for the DEGs associated with the intake main effect.** Genes with all gray cell indicate those that were excluded because they were also significant for the gain by intake interaction term. Jackknife 1 P-value gives the adjusted P-value for the meta-analysis with Cohort 1 removed, Jackknife 2 P-value gives the adjusted P-value for the meta-analysis with Cohort 2 removed, and so on. Yellow cells indicate jackknife analyses where the P-value was insignificant, i.e. the gene failed to pass the jackknife analysis.
(PDF)

**S6 Table. Jackknife sensitivity analysis results for the DEGs associated with the gain by intake interaction.** Jackknife 1 P-value gives the adjusted P-value for the meta-analysis with Cohort 1 removed, Jackknife 2 P-value gives the adjusted P-value for the meta-analysis with Cohort 2 removed, and so on. Yellow cells indicate jackknife analyses where the P-value was

insignificant, i.e. the gene failed to pass the jackknife analysis.
(PDF)

## Acknowledgments

The authors wish to acknowledge Linda Flathman for her outstanding technical and laboratory assistance; the USMARC Core Laboratory for performing the sequencing; Troy Gramke and Sam Nejezchleb for their assistance with sample collection; Donna Griess for assistance with the manuscript, and the USMARC Cattle Operations and Abattoir staff. The USDA is an equal opportunity provider and employer. Mention of a trade name, proprietary product, or specified equipment does not constitute a guarantee or warranty by the USDA and does not imply approval to the exclusion of other products that may be suitable.

## Author Contributions

**Conceptualization:** Amanda K. Lindholm-Perry.

**Formal analysis:** Brittney N. Keel.

**Methodology:** Brittney N. Keel.

**Project administration:** Amanda K. Lindholm-Perry.

**Resources:** Harvey C. Freetly, William T. Oliver.

**Supervision:** Amanda K. Lindholm-Perry.

**Writing – original draft:** Amanda K. Lindholm-Perry, Brittney N. Keel.

**Writing – review & editing:** Harvey C. Freetly, William T. Oliver, Lea A. Rempel.

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
