## [Decision Letter · Decision Letter 0]

21 Oct 2019

PONE-D-19-25581

Genes associated with body weight gain and feed intake identified by meta-analysis of the mesenteric fat from crossbred beef steers

PLOS ONE

Dear Dr. Lindholm-Perry,

Thank you for submitting your manuscript to PLOS ONE. After careful consideration, we feel that it has merit but does not fully meet PLOS ONE’s publication criteria as it currently stands. Therefore, we invite you to submit a revised version of the manuscript that addresses the points raised during the review process.

We would appreciate receiving your revised manuscript by Dec 05 2019 11:59PM. To enhance the reproducibility of your results, we recommend that if applicable you deposit your laboratory protocols in protocols.io, where a protocol can be assigned its own identifier (DOI) such that it can be cited independently in the future. For instructions see: http://journals.plos.org/plosone/s/submission-guidelines#loc-laboratory-protocols

We look forward to receiving your revised manuscript.

Kind regards,

Juan J Loor

Academic Editor

PLOS ONE

**Journal Requirements:**

Reviewer's Responses to Questions

4.

We note that you are reporting an analysis of a microarray, next-generation sequencing, or deep sequencing data set. PLOS requires that authors comply with field-specific standards for preparation, recording, and deposition of data in repositories appropriate to their field. Please upload these data to a stable, public repository (such as ArrayExpress, Gene Expression Omnibus (GEO), DNA Data Bank of Japan (DDBJ), NCBI GenBank, NCBI Sequence Read Archive, or EMBL Nucleotide Sequence Database (ENA)). In your revised cover letter, please provide the relevant accession numbers that may be used to access these data. For a full list of recommended repositories, see http://journals.plos.org/plosone/s/data-availability#loc-omics or http://journals.plos.org/plosone/s/data-availability#loc-sequencing.

**Comments to the Author**

1. Is the manuscript technically sound, and do the data support the conclusions?

Reviewer #1: Yes

Reviewer #2: Yes

Reviewer #3: Yes

2. Has the statistical analysis been performed appropriately and rigorously? 

Reviewer #1: Yes

Reviewer #2: Yes

Reviewer #3: Yes

3. Have the authors made all data underlying the findings in their manuscript fully available?

Reviewer #1: Yes

Reviewer #2: Yes

Reviewer #3: Yes

4. Is the manuscript presented in an intelligible fashion and written in standard English?

Reviewer #1: Yes

Reviewer #2: Yes

Reviewer #3: Yes

5. Review Comments to the Author

Reviewer #1: Dear Authors:

This MS finally made it clear to me that my approach to genomics in the early 2000 s with 5 animals per treatment group and single marker genes for lipogenesis, fat oxidation etc gave us little information. On the other hand this work also did not pinpoint too many DE genes one would think of from a biological-function perspective.

This was a big project and authors and ever new emerging bioinformatic tools allowed you to do extensive analyses and assessment. The overall assessments are interesting since there are other data sets point to CNS, immune function system/inflammation etc. Once again I see no DE for genes involved in protein turnover and of course protein synthesis is under transnational control. There are also many other factors that effect gene expression qualitatively and quantitatively. Such would include epi-genetic effects, nucleosomes and nucleic acid post transcriptional remodeling, mi RNA, long non-coding RNA etc.

I like the way the paper is written, but it may be a bit wordy and all those gene abbreviations do not make for pleasant reading (there appear to be no alternatives either).

I have a few comments/queries:

Line 54 High fat diets various various biological reasons in ruminants seldom exceed 5% fat and in comparison to chickens and pigs such diet could easily contain 20% fat while in humans in the US many consume a 50% plus fat calorie diet. So this high fat is only applicable to ruminants.

Line 112 says: Length of time between the feeding trial end and slaughter were 12-18 etc dates. Gene expression can be pretty sensitive to type and amount of feed. Once the trial was ended, were the animals continued on the same diet and at the same intake level until tissue sampling? Was feed efficiency maintained post feeding trial until slaughter. Would a time course sampling strategy at intervals throughout the feeding trail result more representative data. Often when a trial is over and all animals are to the same finish and weight there are no more differences. That is a problem of sampling only at the end of the trial.

Much of the phenotypic expression that is observed also depends on regulation of enzyme catalysis by covalent modifications and allosteric factors.

Reviewer #2: Manuscript PONE-D-19-25581 aimed to identify genes that may be differentially expressed in steers with high and low BW gain and feed intake. The study is technically and scientifically sound but authors need to improve the main messages that can be taken from their nice research scope.

Line 52 any idea on how much a farmer can save? Please give an estimate in %

Line 53 what is usable? Please elaborate

Lines 68-75 please describe your study hypothesis. It seems that it is somehow encrypted in this paragraph, could you please be more precise on your research expectations

Line 89 is it 64 to 92 days. Is that the sampling period? Please clarify

Lines 88-94 please provide forage to concentrate ratio and the rationale behind the diet formulation i.e., NRC… here and later for the other diet descriptions

Line 91 after steakmater add (protein supplement; Land..

Line 122 add city and country and also catalogue number here and anywhere appropriate

Line 278 how can you overcome the fact that you have 5 cohort animals – does that influence or not your results? Please elaborate

Lines 417 to 425 what would be the take-home message? Please try to deliver more precise. You have 5 cohorts of animals, each of them with precise feeding managements can you relate those variables with your transcriptomic results and BW gain and feed intake. The main issue perhaps is that in its present version your conclusion statements have not been landed in a way that the answered your initial objectives.

Reviewer #3: This is a very well-written manuscript describing a meta-analysis of RNAseq in mesenteric fat tissue in regards to body weight gain and feed efficiency. While I agree with the choice of using an adjusted p-value of 0.01, I would appreciate an acknowledgement of the false positive errors that the lack of concordance between cohorts can be due to genes being identified as DE by chance.

6. PLOS authors have the option to publish the peer review history of their article (what does this mean?). If published, this will include your full peer review and any attached files.

Reviewer #1: No

Reviewer #2: No

Reviewer #3: No

---

## [Author Response · Author response to Decision Letter 0]

26 Nov 2019

Dear Dr. Loor,

We want to sincerely thank you for the opportunity to revise our manuscript for potential publication in PLOS ONE. We are also grateful to the three peer-reviewers for their comments, questions and suggestions. The comments and questions that we received from the peer-reviewers were excellent points and as such, we have attempted to address each one and edit the manuscript accordingly. With the incorporation of these suggestions, we believe the manuscript has been substantially improved.

Sincerely,

Amanda Lindholm-Perry, Ph.D.

Reviewer's Responses to Questions

Response: We have modified our manuscript according to the PLOS ONE guidelines.

Response: We have altered the sentence on Lines 340-342 to include a list of the three differentially expressed heat shock proteins and removed reference to “data not shown”.

Response: We have included a caption for each file of Supporting information that is denoted by bold type, and have chosen to leave the legend.

4. We note that you are reporting an analysis of a microarray, next-generation sequencing, or deep sequencing data set. PLOS requires that authors comply with field-specific standards for preparation, recording, and deposition of data in repositories appropriate to their field. Please upload these data to a stable, public repository (such as ArrayExpress, Gene Expression Omnibus (GEO), DNA Data Bank of Japan (DDBJ), NCBI GenBank, NCBI Sequence Read Archive, or EMBL Nucleotide Sequence Database (ENA)). In your revised cover letter, please provide the relevant accession numbers that may be used to access these data. For a full list of recommended repositories, see http://journals.plos.org/plosone/s/data-availability#loc-omics or http://journals.plos.org/plosone/s/data-availability#loc-sequencing.

Response: Our sequencing data has been uploaded to the NCBI Sequence Read Archive and is accessible under accession number PRJNA528040. This is shown on Lines 149-150.

Reviewer #1: 

This MS finally made it clear to me that my approach to genomics in the early 2000 s with 5 animals per treatment group and single marker genes for lipogenesis, fat oxidation etc gave us little information. On the other hand this work also did not pinpoint too many DE genes one would think of from a biological-function perspective.

Response: We appreciate this comment. Our lab has previously performed several microarray and RNA-Seq experiments using one group of animals and then have gone to validate the genes identified in a separate population of animals. We have been surprised with how few genes overlap. Our thoughts were that a meta-analysis may help us identify genes that are involved in these phenotypes in more than just one group of animals. It is unfortunate that we did not detect more genes with biological functions that we recognized, but not altogether surprising considering that we are evaluating the entire transcriptome rather than using a candidate gene approach. We also believe that not all of the functions of the genes expressed are known. The key really is to test the genes we have identified in an unrelated population to determine whether they hold up. This is our next step.

This was a big project and authors and ever new emerging bioinformatic tools allowed you to do extensive analyses and assessment. The overall assessments are interesting since there are other data sets point to CNS, immune function system/inflammation etc. Once again I see no DE for genes involved in protein turnover and of course protein synthesis is under transnational control. There are also many other factors that effect gene expression qualitatively and quantitatively. Such would include epi-genetic effects, nucleosomes and nucleic acid post transcriptional remodeling, mi RNA, long non-coding RNA etc.

Response: The only genes that have a role in protein turnover that we consistently identified among multiple cohorts were heat shock proteins. We did, however, identify many genes involved in lipid metabolism, which was something that we had expected to see. However, we do agree with this comment and have included information in the discussion regarding other factors that could alter gene expression. Please see Lines 303-308.

Line 54 High fat diets various various biological reasons in ruminants seldom exceed 5% fat and in comparison to chickens and pigs such diet could easily contain 20% fat while in humans in the US many consume a 50% plus fat calorie diet. So this high fat is only applicable to ruminants.

Response: We agree that high fat diets in ruminants are lower than in humans and other species. However, these diets, which are high in fat for ruminants, have been shown to increase mesenteric fat accumulation.

Line 112 says: Length of time between the feeding trial end and slaughter were 12-18 etc dates. Gene expression can be pretty sensitive to type and amount of feed. Once the trial was ended, were the animals continued on the same diet and at the same intake level until tissue sampling? Was feed efficiency maintained post feeding trial until slaughter. Would a time course sampling strategy at intervals throughout the feeding trail result more representative data. Often when a trial is over and all animals are to the same finish and weight there are no more differences. That is a problem of sampling only at the end of the trial.

Response: These are notable observations and we fully agree with the reviewer. Unfortunately, we do not have the data regarding feed efficiency until the end of the study, and we do not have the means to take a time course sample on a tissue like mesenteric fat until slaughter. We do maintain the steers on the same ad libitum feed study ration from the end of the study when we evaluate gain and feed intake through the 12-18 days up to slaughter. While these animals did finish the study on the same date, they do have variation in their intakes and gains even through the end of the study. This information can be viewed in our previous manuscript: Keel BN, Zarek CM, Keele JW, Kuehn LA, Snelling WM, Oliver WT, Freetly HC, and Lindholm-Perry AK. RNA-Seq meta-analysis identifies genes in skeletal muscle associated with gain and intake across a multi-season study of crossbred beef steers. BMC Genom. 2018; 19:430. We are moving to the direction of sampling tissues that are less invasive by biopsy during a feeding trial, unfortunately, we do not think that mesenteric fat will be possible to sample by biopsy.

Much of the phenotypic expression that is observed also depends on regulation of enzyme catalysis by covalent modifications and allosteric factors.

Response: We agree with the reviewer and recognize that this is a limitation of the transcriptomic research. We have added information to the Discussion to address this. Please see Lines 303-308.

Reviewer #2: Manuscript PONE-D-19-25581 aimed to identify genes that may be differentially expressed in steers with high and low BW gain and feed intake. The study is technically and scientifically sound but authors need to improve the main messages that can be taken from their nice research scope.

Line 52 any idea on how much a farmer can save? Please give an estimate in %

Response: This is a great question and I can appreciate the interest in trying to quantify the potential cost savings to the producer. However, this is a challenging request because the actual savings due to improved feed efficiency are going to be specific to the producer and depend on the environment and available feed resources. Unfortunately, we do not think there is a way to address this well. 

Line 53 what is usable? Please elaborate

Response: This has been changed to “meat products for consumption”, Line 54.

Lines 68-75 please describe your study hypothesis. It seems that it is somehow encrypted in this paragraph, could you please be more precise on your research expectations

Response: We have added additional information to the Introduction to better describe our hypothesis. Please see lines 77-82. 

Line 89 is it 64 to 92 days. Is that the sampling period? Please clarify

Response: We have revised this section by reordering sentences to better clarify the study, see Lines 104-124. Animals were on the feeding trial for a total of 64-92 days. At the end of the study, average daily gain and feed intake phenotypes were calculated for all the animals on the trail. A total of 16 animals were selected from each cohort. Those animals were then comingled and allowed to consume the same study ration ad libitum until slaughter, which occurred 11 to 28 days after the feeding trial ended.

Lines 88-94 please provide forage to concentrate ratio and the rationale behind the diet formulation i.e., NRC… here and later for the other diet descriptions

Response: Thank you for identifying this oversight. All diets were formulated to meet or exceed nutrient requirements to express maximum growth potential (NRC 2000). This sentence has been added to the manuscript at Lines 104-105. We have included the exact ration that animals were fed, rather than the ratio of forage to concentrate. With the information provided the ration can be calculated.

Line 91 after steakmaker add (protein supplement; Land..

Response: This has been included on Line 98.

Line 122 add city and country and also catalogue number here and anywhere appropriate

Response: Catalog #, City and country have been added to the Methods.

Line 278 how can you overcome the fact that you have 5 cohort animals – does that influence or not your results? Please elaborate

Response: We do believe that having 5 cohorts of animals influences our results and have previously published the meta-analysis procedure as a method to handle multiple cohorts.

Lines 417 to 425 what would be the take-home message? Please try to deliver more precise. You have 5 cohorts of animals, each of them with precise feeding managements can you relate those variables with your transcriptomic results and BW gain and feed intake. The main issue perhaps is that in its present version your conclusion statements have not been landed in a way that the answered your initial objectives.

Response: Thank you for this comment. We agree that the prior version of the manuscript did not deliver an acceptable conclusion. We have moved that paragraph out of the Conclusions and into a paragraph of the Discussion and modified it, Lines 294-308. We have also included a new Conclusions paragraph, Lines 447-454.

Reviewer #3: This is a very well-written manuscript describing a meta-analysis of RNAseq in mesenteric fat tissue in regards to body weight gain and feed efficiency. While I agree with the choice of using an adjusted p-value of 0.01, I would appreciate an acknowledgement of the false positive errors that the lack of concordance between cohorts can be due to genes being identified as DE by chance.

Response: We actually used an adjusted P-value of 0.05 for the meta-analysis. We have included the following sentences in the first paragraph of the Discussion (Lines 290-293) to acknowledge that there are still false positive errors within the genes identified as differentially expressed with the following statements: “This allowed us to identify a several genes among the 80 animals tested that may be involved in gain and feed intake. Although we did correct the meta-analysis P-values for multiple testing and reported the data using PADJ<0.05, this still implies that 5% of the significant tests are false positives.”

---

## [Decision Letter · Decision Letter 1]

13 Dec 2019

Genes associated with body weight gain and feed intake identified by meta-analysis of the mesenteric fat from crossbred beef steers

PONE-D-19-25581R1

Dear Dr. Lindholm-Perry,

We are pleased to inform you that your manuscript has been judged scientifically suitable for publication and will be formally accepted for publication once it complies with all outstanding technical requirements.

With kind regards,

Juan J Loor

Academic Editor

PLOS ONE

Additional Editor Comments (optional):

Reviewers' comments:

Reviewer's Responses to Questions

**Comments to the Author**

1. If the authors have adequately addressed your comments raised in a previous round of review and you feel that this manuscript is now acceptable for publication, you may indicate that here to bypass the “Comments to the Author” section, enter your conflict of interest statement in the “Confidential to Editor” section, and submit your "Accept" recommendation.

Reviewer #1: All comments have been addressed

Reviewer #2: All comments have been addressed

Reviewer #3: All comments have been addressed

2. Is the manuscript technically sound, and do the data support the conclusions?

Reviewer #1: Yes

Reviewer #2: Yes

Reviewer #3: Yes

3. Has the statistical analysis been performed appropriately and rigorously? 

Reviewer #1: Yes

Reviewer #2: Yes

Reviewer #3: Yes

4. Have the authors made all data underlying the findings in their manuscript fully available?

Reviewer #1: Yes

Reviewer #2: Yes

Reviewer #3: Yes

5. Is the manuscript presented in an intelligible fashion and written in standard English?

Reviewer #1: Yes

Reviewer #2: Yes

Reviewer #3: Yes

6. Review Comments to the Author

Reviewer #1: Very good job of revision.

Line 166... developed in (5) was used; give a name of the authors

Line 290 This allowed us to identify a several genes; remove the "a"

Reviewer #2: authors have made all suggested corrections

Reviewer #3: (No Response)

7. PLOS authors have the option to publish the peer review history of their article (what does this mean?). If published, this will include your full peer review and any attached files.

Reviewer #1: No

Reviewer #2: No

Reviewer #3: No

---

## [Editor Report · Acceptance letter]

20 Dec 2019

PONE-D-19-25581R1 

Genes associated with body weight gain and feed intake identified by meta-analysis of the mesenteric fat from crossbred beef steers 

Dear Dr. Lindholm-Perry:

I am pleased to inform you that your manuscript has been deemed suitable for publication in PLOS ONE. Congratulations! Your manuscript is now with our production department. 

With kind regards,

on behalf of

Dr. Juan J Loor 

Academic Editor

PLOS ONE